# A TUG Value Longer Than 11 s Predicts Fall Risk at 6-Month in Individuals with COPD

**DOI:** 10.3390/jcm8101752

**Published:** 2019-10-22

**Authors:** Vivien Reynaud, Daniela Muti, Bruno Pereira, Annick Greil, Denis Caillaud, Ruddy Richard, Emmanuel Coudeyre, Frédéric Costes

**Affiliations:** 1Service de Médecine du Sport et des Explorations Fonctionnelles, CHU Clermont Ferrand, 63003 Clermont Ferrand, France; vivien.reynaud@gmail.com (V.R.);; 2Service de Médecine Physique et Réadaptation, CHU Clermont Ferrand, 63003 Clermont Ferrand, France; 3Service de Pneumologie, CHU Clermont Ferrand, 63003 Clermont Ferrand, France; 4Clinique Cardio Pneumologique, Durtol, France; 5Service de Biostatistiques, CHU Clermont Ferrand, 63003 Clermont Ferrand, France; 6Centre de Recherche en Nutrition Humaine Auvergne, 63003 Clermont Ferrand, France; 7Université Clermont Auvergne, INRA, UNH, 63000 Clermont Ferrand, France

**Keywords:** Chronic obstructive pulmonary disease, Timed Up and Go test, Postural balance, Fall, Risk factor, Hypoxia

## Abstract

Risk of a fall is increased in individuals with chronic obstructive pulmonary disease (COPD), and is usually evaluated using the Berg Balance Scale (BBS), but this is difficult to perform in everyday clinical practice. We aimed to prospectively predict short-term fall recurrence in COPD patients using a predetermined cut-off value of the Timed Up and Go test (TUG). In stable COPD patients, we collected self-reported records of the number of falls in the previous year, and measured TUG and BBS scores for each individual. Records of fall recurrence were obtained prospectively at 6-months after the initial evaluation. Among the 50 patients recruited, 23 (46%) had at least one fall during the past year. The optimal diagnosis value for the TUG to detect a fall was 10.9 s with a sensitivity of 100% and a specificity of 97%. A cut-off of 11 s predicted fall recurrence with high sensitivity and specificity (93% and 74%, respectively). The TUG as well as the BBS score detected fallers, and a cut-off value of 11 s predicted fall recurrence. TUG could be easily incorporated into the scheduled functional evaluations of COPD patients, could predict the risk of a fall and when appropriate, could guide specific balance training exercises to prevent fall.

## 1. Introduction

Chronic obstructive pulmonary disease (COPD) is an important cause of morbidity and mortality worldwide, impacting daily activities and quality of life. Muscle dysfunction, a frequent comorbidity in individuals with COPD [1], is recognized as one of the risk factors for falls in the elderly. Consequently, fall risk has recently emerged as a concern in COPD. Roig et al. [2] found an annual rate of 1.2 falls per COPD patient. In a large cohort of patients from a UK general practice database, 24% of COPD patients had a fall record, compared to 12% of non-COPD age-matched subjects [3]. In more-severe COPD patients, falls affected up to 50% of patients entering a pulmonary rehabilitation (PR) program [4], and this value was even higher in patients treated with long-term oxygen therapy (LTOT). Importantly, a history of falls in the previous 6 months is associated with a three-fold increased risk of mortality [5].

Falls are caused by impaired balance control, particularly the biomechanics, transition, and gait subcomponents of balance [6]. In a systematic review, Oliveira et al. [7] recommended the Berg Balance Scale (BBS), the Balance Evaluation System test (BEST) and the Activities-specific Balance Confidence score (ABC-score) to assess balance in COPD patients. In COPD patients, a lower BBS score is associated with increased risk of fall and is correlated with the severity of bronchial obstruction [8]. Moreover, a difference of 3.5 points (95% confidence interval: −7.0 to −0.2) could separate fallers from non-fallers [4], and BBS score is responsive to specific balance training incorporated into a PR program [9]. However, BBS requires approximately 20 min to perform, precluding its utilization in a large population and during follow-up visits. In contrast, the Timed Up and Go test (TUG) [10] is a simple, easy to implement, and reproducible test in COPD [11]. Cut-off values of 12 s, proposed to differentiate fallers from non-fallers in community-dwelling elders, showed good sensitivity [12]. This threshold has also been demonstrated in pathologies such as Parkinson’s disease, amyotrophic lateral sclerosis, and post-stroke or orthopedic disturbance [13]. In COPD, the TUG score was also about 3 s longer in patients who had fallen [4,8]. Similarly, Al Haddad et al. [14] found retrospectively that a cut-off value of 12 s predicted the occurrence of at least one fall during the previous year in COPD patients, with a sensitivity and specificity of 74%. Those with a longer TUG score had higher body mass indexes and Medical Research Council (MRC) dyspnea scores, and a shorter, 6-min walking distance. After validation in the elderly and in neurological disorders, compelling evidence showed clinical interest in TUG as a screening assessment of the risk of falling in COPD patients. However, the sensitivity of TUG to discriminate fallers has never been compared to the reference BBS test, and the proposed TUG threshold values have not been validated prospectively in COPD.

Our main aim was thus to prospectively establish a cut-off value of TUG to predict falls in the short term. The secondary aim was to assess consistency of TUG scores relative to BBS scores for risk of fall screening in COPD patients. Finally, we compared patients with or without chronic respiratory failure to assess the role of chronic hypoxia.

## 2. Materials and Methods

### 2.1. Patients

We recruited outpatients consulting for scheduled COPD follow-up in a respiratory medicine department. The diagnosis of COPD was made according to the GOLD recommendations on non-reversible bronchial obstruction using spirometry (with a post bronchodilator FEV1/FVC lower than 0.7). All the patients gave their written informed consent, and the study was approved by our local ethics committee (CPP Sud Est 6 2016-A01188-43).

We included patients who had been free of exacerbation in the previous 4 weeks, >3 months after completion of a respiratory rehabilitation program. Patients older than 80 years were excluded in order to limit the well-known influence of age on postural balance. In order to obtain patients with variable severities of disease, we included an equal number of COPD patients with chronic respiratory failure who were receiving long-term oxygen therapy (LTOT+ group), and patients without hypoxemia (LTOT− group). All the patients treated with LTOT presented an arterial oxygen pressure below 55 mHg when LTOT was prescribed but blood gases were not controlled at the time of the study. We excluded patients with a history of stroke or neurological disorder, unstable heart failure, or lower limb locomotor impairment. Inclusion and non-inclusion criteria were screened by studying the medical file then carrying out an interrogation (about medical history in particular) and an extensive clinical examination.

### 2.2. Methods

All evaluations and tests were performed by one investigator (VR) on the same day and in the same order. Resting periods were respected between walk tests and muscle strength measurements.

### 2.3. Patient Characteristics


Pulmonary volumes, bronchial airflows, and carbon monoxide lung diffusing capacity (DLCO) were measured in a body plethysmograph (Sentrysuite, Viasys, Conshohocken, PA, USA), according to European Respiratory Society recommendations [15]. All the measurements were compared with the European Community for Coal and Steel (ECCS) predicted values [16]Six-minute walking distance (6MWD) was measured according to international recommendations [17]. In patients treated with LTOT, the test was performed with O2 flow prescribed for walking.Body composition was determined using biphotonic absorptiometry (Hologic QDR-2000, software version V5.67A, Hologic Inc., Bedford, MA), with a single scan mode [18,19]. We determined fat mass (FM) and fat-free mass (FFM). We calculated the FFM index (FFMI, the ratio of FFM over height^2^, kg m^−2^) and skeletal mass index (SMI, the ratio of appendicular FFM over height^2^, kg m^−2^).Maximal isokinetic concentric strength of quadriceps was measured at 60°/s (Cybex Norm, USA). Evaluation was performed in a sitting position according to the manufacturer’s instructions. Range of motion was set from 90° to 0° of knee flexion. Before each test, a gravity compensation procedure was performed. The test consisted of two repetitions for habituation followed after a resting period by four repetitions at 60°/s in a concentric mode. The best result was retained as the peak torque (PT) value and was normalized to patients’ weight.Anxiety and depression were evaluated by the Hospital Anxiety and Depression Scale (HADS).


### 2.4. Balance Assessment and Fall Records


Timed Up and Go Test (TUG) [10]. Subjects were instructed to stand up from a standard armchair at the word “go”, walk at a regular pace for 3 m (indicated by a line on the floor), turn around, walk back, and sit down. This assesses lower limb muscle force, gait speed, and coordination—three components of postural control. TUG shows excellent intra- and inter-observer reliability, with an intraclass correlation coefficient of 0.99 [20]. A practice test was followed by the actual test, with an opportunity to recover between the trials. Gait aids were permitted when appropriate and, for patients treated with LTOT, the test was performed with O_2_ flow prescribed for walking that was carried in a shoulder bag or backpack. A time longer than 12 s is considered as an indicator of fall risk in the elderly and in COPD patients [14,21].The Berg Balance Scale [22] was performed by the same investigator (VR). It comprised 14 questions and tasks completed by patients and scored by the observer, from worst (0) to best (4). The maximum total score is 56 points, and the risk of fall is considered low above 45.The number of falls in the previous year was collected with the Elderly Fall Screening Test questionnaire [23]. A fall was defined as a positive response to the question “Have you ever found yourself on the ground without wanting to get there, while sitting, standing or lying down?” We considered patients as fallers with the occurrence of at least one fall.Fall recurrence at 6 months after evaluation was prospectively obtained via a phone survey using a standardized questionnaire with the same definition as previously used. During this phone survey, we also assessed whether a rehabilitation program was undertaken during the observational period.


### 2.5. Statistical Analysis

We estimated that 40 patients per group were necessary to obtain a BBS score difference of 3.5 ± 5.0 points between fallers and non-fallers for a two-sided type I error at 5% and a statistical power greater than 80% [4]. An interim analysis was planned in protocol and carried out when half the inclusions were recruited (i.e., for 20 patients per group). According to (1) these results highlighting an effect size at 2.77 (95% CI; 1.89; 3.64) and (2) the type I spending function, it was proposed that we limit the study to 25 patients per group to guarantee a satisfactory statistical power of secondary endpoints.

All analyses were performed with Stata software (Version 13, StataCorp, College Station, TX, US). The level of significance was set for a type I error at 5%. Data are presented as mean and standard deviations. The Shapiro–Wilk’s test was used to test for normality. Concordance of abnormal tests between BBS and TUG was calculated using Cohen’s kappa coefficient. For abnormality, we used a cut-off value of 45 for BBS and 12 s for TUG. Sensitivity and specificity values were estimated.

A receiver operating characteristic (ROC) curve analysis was carried out and the best threshold value of TUG to detect risk of fall was estimated according to clinical relevance and indexes reported in the literature. The accuracy of this threshold was assessed by ROC curve analyses considering the recurrence of falls during the follow-up (requiring at least one fall). For comparisons of continuous variables between fallers and non-fallers, and between LTOT+ and LTOT−, we used Student’s *t* tests, or Mann–Whitney U-tests if assumptions of the *t*-test were not met (normality and homoscedasticity). Categorical variables were compared between groups using Chi-squared or Fisher’s exact tests. Multivariable analysis was then performed using generalized linear models (logistic regression for binary endpoints of (i) past falls and (ii) future falls (6 months after initial evaluation), which were dichotomized as follows: 0 vs ≥1) to take into account confounder variables. The covariates were determined using univariate results and clinical relevance (age, gender, BMI, 6MWD, and previous falls). Particular attention was paid to the study of multicollinearity and interactions between covariates (1) studying the relationships between the covariables and (2) evaluating the impact of the addition or deletion of variables on multivariable models. The results were expressed as predicted probabilities and 95% confidence intervals.

## 3. Results

### 3.1. Patient Characteristics

We included 50 COPD patients, 25 each in the LTOT+ and LTOT− groups, respectively. Twenty-three (46%) patients fell in the previous year (mean number of falls per patient: 2.3 ± 3.9), with a significantly higher number in the LTOT+ than in the LTOT− group (17 (68%) vs 6 (24%), respectively, *p* < 0.005). Patient characteristics, pulmonary function, and body composition compared between the two groups and in relation to the incidence of fall are detailed in Table 1 and Table 2. In the whole population, HAD scores were on average within normal values but with a large range: 6.5 ± 4.1 (range 0–16) and 5.6 ± 4.0 (range 0–18) points for anxiety and depression, respectively. These scores were not affected by the LTOT group and faller/non-faller groups.

### 3.2. BBS, TUG, and Fall

BBS score was consistently lower, and TUG value was longer, in the faller group than in non-fallers (40.5 ± 6.8 points vs 54.8 ± 1.8 points, and 15.4 ± 3.9 s vs 7.5 ± 1.4 s, respectively; both *p* < 0.001). There was no overlap in the standard deviation of TUG values between fallers and non-fallers. Using the 12-s cut-off value [14], agreement between TUG and BBS for detection of fallers was excellent, with a κ coefficient of 0.92 (*p* < 0.0001) in the whole population (Table 3). The sensitivity and specificity of TUG were 95% and 97%, respectively. The area under the ROC curve was 0.99 (95% confidence interval; 0.96; 1.00), giving the optimal diagnostic value of TUG at 10.9 s to detect falls (Figure 1). Repeating the analysis with the 11-s cut-off value, we confirmed the excellent agreement (Table 3). This threshold of 11 s for TUG nicely predicted fall incidence 6 months after the initial evaluation (Area Under the Curve, AUC 0.87 (0.77–0.97)); sensitivity and specificity were 93.3% (68.1; 99.8) and 74.2% (55.4; 88.1), respectively. Multivariable analysis adjusted for age (*p* = 0.81), gender (*p* = 0.45), BMI (*p* = 0.20), 6MWD (*p* = 0.84), previous falls (*p* = 0.50), and LTOT (*p* = 0.96) confirmed that TUG remained a robust predictor of future falls with a predicted probability of 0.59 (0.50; 0.67) (*p* = 0.045).

### 3.3. Predictive Factors Associated with Fall

Fallers presented with significantly lower Forced Vital Capacity (FVC) and Total Lung Capacity (TLC) values in the LTOT+ group, and there was a lower DLCO value in the LTOT− group (Table 1). In the LTOT− group, FFMI was lower in fallers and SMI was surprisingly higher (Table 2). The latter was also increased in fallers of the LTOT+ group. In the whole population, fallers had significantly lower BMI (*p* = 0.04), FFM (*p* = 0.02), FFMI (*p* = 0.006), and lower limb FFM (*p* = 0.02) compared with the non-fallers. Maximal isokinetic quadriceps strength and 6MWD were also lower in fallers.

LTOT treatment significantly affected lung function, with the FEV1/FVC ratio and DLCO higher in the LTOT− than the LTOT+ group (51.6 ± 13.6 vs. 42.3 ± 11.4%, *p* = 0.01; and 52.8 ± 19.2 vs. 36.9 ± 13.1% predicted, *p* = 0.005; for the two measures, respectively). Body composition likewise differed between groups, with BMI and percentage of fat mass lower in the LTOT− group (25.6 ± 5.1 vs. 30.4 ± 6.8 kg·m^−2^, *p* = 0.007; and 25.0 ± 7.8 vs. 30.7 ± 7.1%, *p* = 0.01; for the two measures respectively). Whole body fat-free mass and appendicular FFM were similar in the two groups.

Multivariable analysis of predicted factors of past falls is presented in Figure 2. Adjusted for patients’ gender (*p* = 0.64), age (*p* = 0.25), and BMI (*p* = 0.15), 6MWD expressed in percentage of the predicted value was the sole significant determinant of fall in our study population (predicted probability = 0.48 (0.46; 0.49), *p* = 0.005).

## 4. Discussion

The main result of our study was the excellent prediction of likelihood of falling in the subsequent 6 months using a cut-off value of 11 s to complete the TUG, with a recurrence of fall in 62% of patients in whom TUG was longer than this threshold. Moreover, we found an excellent agreement between the Timed Up and Go test value and the BBS score for detection of fall risk in COPD patients who had allowed us to determine the threshold value.

### 4.1. TUG as a Screening Test for Fall Risk

Our study was designed to validate use of TUG as a replacement for the BBS in detecting risk of fall. A significantly slower time to perform TUG was previously reported in fallers compared to non-fallers [4,14], and a cut-off value of 12 s was proposed to detect risk of fall with a sensitivity and specificity of 74%. We built on these results to show that the predictive value of TUG was as high as that of the BBS score, which remains the gold standard in assessing balance control. The search for simple field tests to assess muscle function and exercise capacity in COPD patients has recently intensified. Thus, TUG may serve the double purpose of detecting fall risk while evaluating locomotor function.

To the best of our knowledge, TUG has never been used prospectively to predict fall risk in COPD patients. We found that the cut-off value of 11 s precisely separates fallers and non-fallers, and predicts relapse of fall during the 6 following months. We acknowledge that our follow-up was of a short duration, but when a high risk of falling is detected then balance training should be rapidly initiated [24], so our results are relevant to the design of optimal treatment regimens.

The between-day 95% confidence interval of minimal detectable change in TUG was shown to be 3 s [25], and the variation after pulmonary rehabilitation ranged from 0.9 to 1.4 s [11]. The difference that we report between fallers and non-fallers largely exceeded the intrinsic variation of TUG, indicating consistency and reinforcing the clinical utility of this test. In the LTOT+ group, carrying the O2 canister could increase the time taken to perform TUG, but we found a similar agreement between BBS and TUG in the LTOT+ or LTOT− groups, which further highlights the consistency of the 11-s cut-off value. TUG value is strongly correlated with both isometric and isokinetic quadriceps PT [26], with 6MWD [11], and also with the BODE index [14]. The response of TUG to pulmonary rehabilitation and/or specific balance control training is known. Consequently, TUG could be used to assess the benefit of such training programs for reducing fall risk. However, we acknowledge that TUG did not discriminate between the different components of postural balance, and a more comprehensive test, such as BBS, should be used for an in-depth assessment.

### 4.2. Fall Incidence in COPD Patients

Although our patient sample was small, we confirmed the high prevalence of fall in COPD patients in stable condition. Our study participants exhibited less severe bronchial obstruction and were younger than those in the previous studies of Beauchamp et al. [4] and Roig et al. [2], but were similar to those of Al Haddad et al. [14]. Only one of our study participants had been enrolled in a rehabilitation program in the year prior to this study, and none presented a recent exacerbation. This is important because acute exacerbation of COPD alters balance control test results relative to those of patients in a steady state [8]. This demonstrates that patients’ balance control should be assessed when they are stable. The onset of fall was two-fold higher in the LTOT+ group, but it was out of the scope of this study to determine whether this higher prevalence was due to the severity of the disease or the effect of chronic hypoxia on postural control. The two LTOT groups differed in COPD severity and BMI but not in muscle mass, which might account for an altered postural control. Moreover, when patients were interviewed by phone 6 months after the initial evaluation, we recorded another fall in 87% of the faller group, and only in 7% of the non-faller group. This confirms the ongoing elevated risk of fall in this population.

### 4.3. Predictive Factors Associated with Falling

After adjusting for age, gender, and BMI, 6MWD (as a percentage of predicted value) remained the only significant predictor of fall in our patients (Figure 2). However, the specific sensitivity of 6MWD is unknown. Moreover, in COPD patients, walking distance is limited by muscle function and respiratory capacity, and it is unlikely that 6MWD could assess the benefit of specific balance training exercises as well as TUG. Age, gender, oxygen saturation, FEV1, and TUG predicted fall risk in COPD patients [14] with similar ages and FEV1 values, but with lower mean 6MWDs than our study patients. We found a consistently lower 6MWD and isokinetic quadriceps PT in fallers compared with non-fallers, but BMI and appendicular muscle mass tended to be higher in the latter. This unexpected result may be explained by selection bias, with a high number of obese COPD patients in our LTOT+ group. These patients are known to present with a larger muscle mass than thin patients. Thus, alteration of postural control due to excessive weight and associated locomotor limitations would be detected earlier than muscle wasting. This point deserves further investigation.

### 4.4. Limitations of the Study

We acknowledge that the number of patients studied was limited. However, we found a similar prevalence of fallers in previous studies, indicating that our population was not biased according to this outcome. We also aimed to compare patients with and without LTOT to confirm a higher prevalence of fall in the LTOT+ group, and the present study was designed to include equal numbers of patients in both groups, which is not representative of the COPD population Using power analyses based on previously published data, we calculated that 40 patients per group were sufficient to test the agreement between BBS and TUG (according to Cohen’s test). As scheduled in the statistical analysis plan, we performed an intermediary analysis that confirmed our hypothesis with a sample of 25 patients. This further emphasizes the power of TUG to detect fallers. However, the small number of patients could limit multivariable regression analyses and explain why known determinants of falling such as muscle mass/strength were not significant in the present study. Nonetheless, we included in the multivariable models only clinically consistent factors (three to five predictive factors) and we feel confident that our results are powerful and robust.

Fall occurrence was self-reported, as is common in similar studies, and we did not control for the consequences of fall. Fall occurrence is often underestimated by patients unless it carries health consequences such as injuries, emergency visits, or hospitalizations. We thus feel confident that, if anything, we underestimated the number of falls. The study was proposed to consecutive patients and regularly followed up in our respiratory unit, but we cannot rule out the possibility that participants were fearful of falling. Once again, we found a similar prevalence in other studies and we feel confident that our recruitment was not unusually biased.

Finally, we did not assess the reproducibility of TUG, but the within-day variation of TUG in COPD patients did not exceed 1 s on average, far lower than the difference reported between fallers and non-fallers [27].

## 5. Conclusions

In individuals with COPD studied in stable respiratory condition, a cut-off value of 11 s predicts the recurrence of falls in the next 6 months with sensitivity exceeding 90% and specificity at 74%. The TUG shows excellent agreement with the Berg Balance Scale score for detecting the onset of fall in the preceding year. We conclude that TUG could be easily incorporated into the scheduled functional evaluations of COPD patients to prevent risk of fall and, when appropriate, to guide specific balance training exercises.

## Figures and Tables

**Figure 1 jcm-08-01752-f001:**
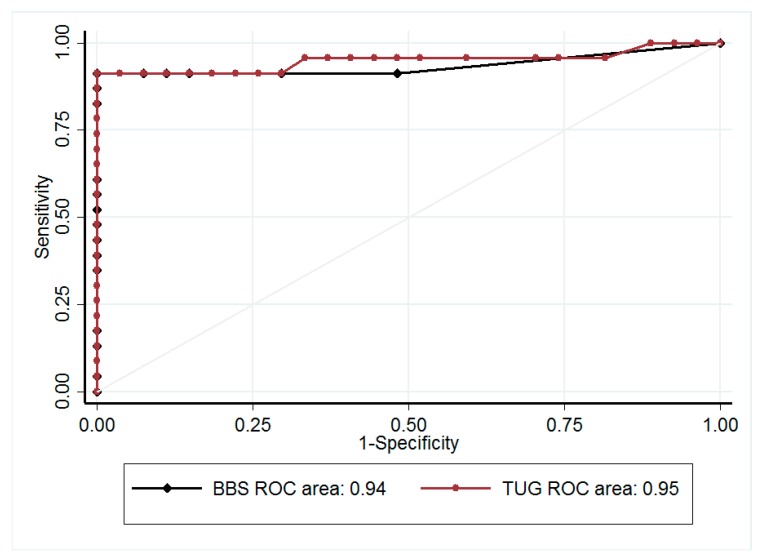
Receiver operating characteristic (ROC) curves for detecting fallers with the Berg Balance Scale score (BBS, black) and with the Timed Up and Go test (TUG, red). Area under curves (AUC) is indicated at the bottom of the paragraph.

**Figure 2 jcm-08-01752-f002:**
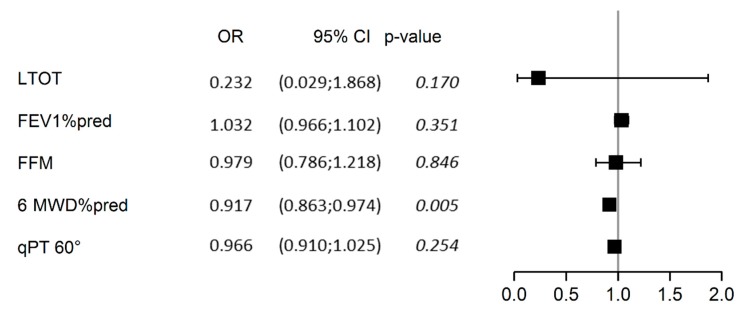
Forest plot of the multivariable analysis of determinants of past falls. The analysis was adjusted for gender, age, and BMI, all of which were non-significant determinants, and which were therefore omitted from the graph for clarity. LTOT, treatment with long-term oxygen therapy; FFM, fat-free mass (kg); 6MWD%pred, 6-min walking distance expressed as a percentage of the predicted value; qPT 60°, isokinetic quadriceps peak torque (PT) at concentric 60°.

**Table 1 jcm-08-01752-t001:** Characteristics of study participants and pulmonary function test results in faller and non-faller groups, grouped according to long-term oxygen therapy (LTOT) treatment.

	All	LTOT−			LTOT+			Effect of LTOT *p* Value	
		Fallers	Non-Fallers	*p*	Fallers	Non-Fallers	*p*	In Fallers	In Non-Fallers
***N***	50	6	19		17	8			
**Age (years)**	66.2 ± 8.2	65.0 ± 4.1	64.2 ± 7.3		68.6 ± 9.8	66.4 ± 8.8			
**BMI (kg m^−2^)**	28.0 ± 6.5	24.2 ± 3.8	26.0 ± 5.5		32.0 ± 7.4	27.0 ± 3.7		0.004	
**FEV1 % pred**	51.8 ± 15.9	63.3 ± 20.4	51.4 ± 15.4		50.0 ± 15.4	47.6 ± 13.2			
**FVC % pred**	87.1 ± 20.7	88.3 ± 21.1	83.1 ± 19.8		82.6 ± 15.4	105.4 ± 25.2	0.04		0.04
**TLC % pred**	123.0 ± 21.7	113.7 ± 31.7	122.2 ± 18.8		118.4 ± 18.0	141.8 ± 19.8	0.01		0.03
**RV/TLC %**	55.3 ± 11.6	54.1 ± 8.7	54.1 ± 13.1		57.2 ± 12.0	55.2 ± 9.6			
**DLCO % pred**	45.5 ± 18.3	38.6 ± 7.5	57.5 ± 19.7	0.01	37.6 ± 13.0	33.7 ± 15.9			
**6MWD % pred**	79.0 ± 24.2	71.1 ± 15.8	93.8 ± 12.1	0.02	59.6 ± 26.4	91.0 ± 14.8	0.001	0.03	
**qPT (Nm/kg)**	116.5 ± 37.9	98.1 ± 23.5	134.4 ± 42.3	0.02	95.0 ± 26.3	133.4± 29.6	0.008		
**Use of assistive device (cane)**	2 (4%)	0	0		2 (11.7%)	0			

Data are given as means ± SD; when significant, *p* values are given between groups for the effects of fall and of LTOT). BMI: body mass index; FEV1: Forced Expiratory Volume in 1 s; FVC: Forced Expiratory Volume; TLC: Total Lung Capacity; RV: Residual Volume; DLCO: diffusing capacity for carbon monoxide; 6MWD: 6-min walking distance; qPT: quadriceps peak torque; pred: predicted value.

**Table 2 jcm-08-01752-t002:** Body composition measured by biphotonic absorptiometry in fallers and non-fallers grouped according to long-term oxygen therapy treatment.

	All	LTOT−			LTOT+			Effect of LTOT *p* Value	
		Fallers	Non-Fallers	*p*	Fallers	Non-Fallers	*p*	In Fallers	In Non-Fallers
**FM (%)**	27.8 ± 7.9	28.9 ± 6.2	23.7 ± 8.0		32.5 ± 6.8	26.9 ± 6.5			
**FFM (kg)**	53.1 ± 11.7	44.8 ± 9.8	52.8 ± 12.7		56.6 ± 9.9	52.5 ± 12.6		0.03	
**FFM (%)**	68.8 ± 8.4	68.4 ± 5.8	71.6 ± 10.3		65.0 ± 6.5	70.2 ± 6.2			
**FFMI (kg·m^−2^)**	20.0 ± 3.4	17.3 ± 3.2	19.8 ± 3.1	0.04	21.3 ± 19.7	19.7 ± 3.6		0.007	
**SMI (kg·m^−2^)**	7.7 ± 1.5	6.7 ± 0.9	7.7 ± 1.3		8.1 ± 1.5	7.8 ± 1.7	0.04	0.02	

Data are mean ± SD. LTOT−, patients not treated with long-term oxygen therapy; LTOT+, patients treated with long-term oxygen therapy; FM, fat mass; FFM, fat-free mass; FFMI, fat-free mass index; SMI, skeletal mass index.

**Table 3 jcm-08-01752-t003:** Agreement between Berg Balance Scale and Timed Up and Go test to detect the risk of fall.

	All	LTOT−	LTOT+
**With a cut-off value of 12 s for TUG**			
**κ coefficient**	0.92	0.87	0.92
**Sensitivity (%)**	95.0 (75.1–99.9)	80.0 (28.4–99.5)	100 (78.2–100)
**Specificity (%)**	96.7 (82.8–99.9)	100 (83.0–100)	90.0 (55.5–99.7)
**PPV (%)**	95.0 (75.1–99.9)	100 (39.8–100)	93.8 (69.8–99.8)
**NPV (%)**	96.7 (82.8–99.9)	95.2 (76.2–99.9)	100 (66.4–100)
**With a cut–off value of 11 s for TUG**			
**κ coefficient**	0.96	1.00	0.92
**Sensitivity (%)**	100 (83.2; 100)	100 (47.8; 100)	90.0 (55.5–99.7)
**Specificity (%)**	96.7 (82.8; 99.9)	100 (83.2; 100)	93.8 (69.8–99.8)
**PPV (%)**	95.2 (76.2; 99.9)	100 (47.8; 100)	100 (66.4–100)
**NPV (%)**	100 (88.1; 100)	100 (83.2; 100)	90.0 (55.5–99.7)

Data are given as means and 95% confidence intervals. LTOT−, patients not treated with long-term oxygen therapy; LTOT+, patients treated with long-term oxygen therapy; PPV, positive predictive value; NPV, negative predictive value.

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
