# Peer review of "A TUG Value Longer Than 11 s Predicts Fall Risk at 6-Month in Individuals with COPD"

_jcm, 2019, doi:10.3390/jcm8101752_

Round 1

Reviewer 1 Report

This manuscript has the potential for important clinical implications. The assessment of future falls is a strength but it requires some revisions. Please see below for your consideration:

Introduction

Page 2, line 52 – please move the description of the TUG to the methods section.

Page 2, line 55 – for clarity please refer only to the cut off value for experiencing a fall (12sec).

Page 2, line 59 – please add a specific value to indicate the difference in time taken to perform the TUG between people with COPD who had fallen compared to those who had not

Page 2, line 62 – replace lower 6MWD with shorter 6MWD

Page 2, line 66 – are you referring to the reference values for the BBS? Please clarify.

Methods

Page 2, line 73 – please state how a COPD diagnosis was confirmed.

Page 2, line 76 – Please explain why those more than 80y were excluded.

Page 2, line 80 – why include LTOT v no LTOT? Is this related to a specific aim, if so please specify. Also how was hypoxia assessed (i.e. were all those on LTOT hypoxic?)

Page 3, line 113 – was LTOT carried by patients or pulled on a trolley? This could have affected time taken to perform the TUG. Please consider.

Page 3, line 120 – please include the definition of a fall.

Page 3, line 122 – how was physical activity measured? What was the intervening period and what occurred during this time? Information about physical activity, if assessed, needs to be reported in the results section.

Statistics – Did you consider correcting for previous falls when predicting future falls? I believe this would be important as experiencing a previous fall is known to be a risk factor for future falls.

                 - Please consider applying a penalised model to account for the small sample size (sparse data bias).

Results

- Please consider converting OR to predicted probabilities to enable the results to be expressed in a more meaningful way.

Page 4, line 159 – It states the HADS scores are within the normal range but according to the SD a proportion of the population had at least a probable presence of anxiety/depression. Please re-phrase.

- Please include the p values in table 1.

Discussion

Page 7, line 221 – please include the percentage at which a cut off of 11sec on the TUG can accurately predict future falls in a 6m follow up period.

- There is a little repetition in the introduction and discussion sections. Please revise and remove.

- One criticism of the TUG is that it cannot inform a balance training program as it does not assess different sub-systems of balance. It can however, highlight a need for further assessment/enquiry. Please consider including this point in your discussion section.

- The discussion section could be revised to be more focused. Comments about sarcopenia and other functional tests do not appear to be related to this study.

- If 6MWD predicts falls and most PR programs will include a 6MWT then do we require an additional screening test for falls? Please consider and comment.

Minor comments

- Some references appear written in the text. Please revise.

- Please review for grammar.

Author Response

Responses to reviewer 1

We thank the reviewer for his helpful comments and criticisms. Please find below a point by point response. We changed the manuscript accordingly.

Introduction

Page 2, line 52 – please move the description of the TUG to the methods section.

We moved the description of TUG to the methods section, on line 114, and the paragraph is rewritten as follows: “Subjects were instructed to stand up from a standard armchair on the word “Go”, to walk at a regular pace for 3 m (indicated by a line on the floor), to turn around, to walk back, and to sit down . It assesses lower limb muscle force, gait speed and coordination, 3 components of postural control. TUG shows excellent intra- and inter-observer reliability, with an intraclass correlation coefficient of 0.99 [12]. A practice test was followed by the actual test with an opportunity to recover between the trials. Gait aids were permitted when appropriate and for patients treated with LTOT, the test was performed with O2 flow prescribed for walking. A time longer than 12 s is considered as an indicator of fall risk in the elderly and in COPD patients [15,21]. »

Page 2, line 55 – for clarity please refer only to the cut off value for experiencing a fall (12sec).

Current literature is not so definite regarding the actual cut-off value in different populations and we chose to be more precise in the first version of the manuscript. However, we agree with the reviewer and for clarity we left only the 12 s cut-off value to separate fallers from non fallers in elderly.

Page 2, line 59 – please add a specific value to indicate the difference in time taken to perform the TUG between people with COPD who had fallen compared to those who had not

The difference in time taken to perform TUG was on average 3 s in the 2 references cited. This has been added on line 59: “ In COPD, TUG was also about 3 s longer in patients who had fallen [4,8] ».

Page 2, line 62 – replace lower 6MWD with shorter 6MWD

done

Page 2, line 66 – are you referring to the reference values for the BBS? Please clarify.

 Here, we refer to the TUG threshold value which has not been validated prospectively in COPD. This has been clarified: “However, the sensitivity of TUG to discriminate fallers has never been compared to the reference BBS test, and the proposed TUG threshold values have not been validated prospectively in COPD. “

Methods

Page 2, line 73 – please state how a COPD diagnosis was confirmed.

The diagnosis of COPD was confirmed according to the 2019 GOLD report (https://goldcopd.org/wp-content/uploads/2018/11/GOLD-2019-v1.7-FINAL-14Nov2018-WMS.pdf). The following sentence has been added: “The diagnosis of COPD was made according to GOLD report with non reversible bronchial obstruction on spirometry (post bronchodilator FEV1/FVC lower than 0.7).”

Page 2, line 76 – Please explain why those more than 80y were excluded.

This has been precised as follows: “We included patients free of exacerbation in the previous 4 weeks, >3 months after completion of a respiratory rehabilitation programme. Patients older than 80 years were excluded to limit the well known influence of age on postural balance. »

Page 2, line 80 – why include LTOT v no LTOT? Is this related to a specific aim, if so please specify. Also how was hypoxia assessed (i.e. were all those on LTOT hypoxic?)

The risk of falling was increased in COPD patients treated with LTOT in the study by Beauchamp et al. (reference 4). So, we sought to separate patients with LTOT or not in the design of the present study. Our results confirm this finding. All the patients in the LTOT+ group presented hypoxaemia at the time of LTOT initiation (in accordance with international guidelines).

This has been precised on page 2 line 70 as follows: “Finally, we compared patients with or without chronic respiratory failure to assess the role of chronic hypoxia.”

And line 84: “All the patients treated with LTOT presented an arterial oxygen pressure below 55 mHg when LTOT was prescribed but blood gases were not controlled at the time of the study.”

Page 3, line 113 – was LTOT carried by patients or pulled on a trolley? This could have affected time taken to perform the TUG. Please consider.

All the patients in LTOT+ group carried the canister in a shoulder bag or a backpack. This could obviously affect TUG performance. However, we found an excellent agreement between BBS and TUG to detect the risk of fall using the 11 s cut-off value in both groups (table 3). So we feel confident that the proposed cut-off value could be used in LTOT patients as well.

This has been precised in paragraph 2.4 (page 3 line 124) and discussed (lines 260-263: “In LTOT+ group, carrying on the O2 canister could increase the time taken to perform TUG but we found a similar agreement between BBS and TUG in LTOT + or LTOT- groups, which further highlights the consistency of the 11s cut-off value.”

Page 3, line 120 – please include the definition of a fall.

The definition of a fall has been added on line 131: A fall was defined as a positive response to the question: “have you ever find yourself on the ground without wanting to get there, while sitting, standing or lying down?”

Page 3, line 122 – how was physical activity measured? What was the intervening period and what occurred during this time? Information about physical activity, if assessed, needs to be reported in the results section.

We acknowledge this sentence is unclear. Physical activity was not measured and the phone survey only allows to assess whether a rehabilitation programme had been undertaken meanwhile. Moreover, the term “intervening period” is mistaken. The sentence has been rewritten as follows: “During this phone survey, we also assess whether a rehabilitation programme has been undertaken during the observational period.”

Statistics – Did you consider correcting for previous falls when predicting future falls? I believe this would be important as experiencing a previous fall is known to be a risk factor for future falls.

We thank the reviewer for the interesting comment. If previous falls was added as an adjustment covariate in multivariable analysis, TUG remained a robust predictor of future falls. We modified the sentence on page 5, line 199: “Multivariable analysis adjusted on age (p=0.81), gender (p=0.45), BMI (p=0.20), 6MWD (p=0.84), previous falls (p=0.50) and LTOT (p=0.96) confirmed that TUG remained a robust predictor of future falls with a predicted probability of 0.59 [0.50; 0.67] (p=0.045).”

                  - Please consider applying a penalised model to account for the small sample size (sparse data bias).

We thank the reviewer for the helpful comment. If the sample size is not so high, it still seems sufficient for in order to perform a generalized linear model like logistic regression, especially due to balance between faller and non-fallers. However, according to the reviewer’s comment, we have realized a penalized model (LASSO for Least Absolute Shrinkage and Selection Operator), which confirms that 6MWD expressed in percentage of the predicted value was the sole significant determinant of fall in our study population (p<0.001).

 Results

- Please consider converting OR to predicted probabilities to enable the results to be expressed in a more meaningful way.

Odds ratio have been converted in predicted probabilities in the result section on page 5, line 201 and page 7 line 228.

 Page 4, line 159 – It states the HADS scores are within the normal range but according to the SD a proportion of the population had at least a probable presence of anxiety/depression. Please re-phrase.

These has been rewritten as follows (page 4, line 173) : “In the whole population, HAD scores were on average within normal values but with a large range: 6.5±4.1 [range 0-16] and 5.6±4.0 [range 0-18] points for anxiety and depression respectively. These scores were not affected by LTOT group and faller/non-faller groups.”

- Please include the p values in table 1.

Tables 1 and 2 have been modified to include the actual p value when between groups differences reached significance (otherwise, for clarity p values were not indicated).

 Discussion

Page 7, line 221 – please include the percentage at which a cut off of 11sec on the TUG can accurately predict future falls in a 6m follow up period.

Among the patients with a TUG ≥11 s, 62% (13/21) presented a relapse of fall in the 6 following months. The first sentence of discussion has been added as follows: “The main result of our study is the excellent prediction of the likelihood of falling in the subsequent 6 months using a cut-off value of 11 s to complete TUG, with a recurrence of fall in 62% of patients in whom TUG was longer than this threshold ».

 - There is a little repetition in the introduction and discussion sections. Please revise and remove.

We have assessed for repetition and changed the discussion accordingly (see the point below).

- One criticism of the TUG is that it cannot inform a balance training program as it does not assess different sub-systems of balance. It can however, highlight a need for further assessment/enquiry. Please consider including this point in your discussion section.

We agree with the reviewer regarding the need of a comprehensive assessment of postural balance. Our goal was to validate the use of TUG in the context of COPD, regarding the high prevalence of fall in this population. In the hand of a non-specialized physician, TUG could detect postural balance impairment and the need for further investigation and/or readaptation. The following sentence has been added on line 269: “However, we acknowledge that TUG did not discriminate between the different components of postural balance and a more comprehensive test, such as BBS, should be used for an in-depth assessment .”

 - The discussion section could be revised to be more focused. Comments about sarcopenia and other functional tests do not appear to be related to this study.

As suggested, the discussion has been focused on the strengths and limitations of the study compared to previous data on the field. Speculative considerations or repetition with the introduction section have been removed; specifically, we deleted sentences on lines 246, 20, 252 and 287.

- If 6MWD predicts falls and most PR programs will include a 6MWT then do we require an additional screening test for falls? Please consider and comment.

Though we found that 6MWD predicted falls in the present study, its specific sensitivity to detect fall is not known.  In COPD patients, walking test investigates muscle function, as TUG, but also cardiorespiratory capacity. So, we are not convinced that 6MWD could predict as well as TUG an impaired postural balance. Anyway, further studies are warranted before concluding that 6MWD could replace TUG and could asses the benefit of a specific balance training progamme. The following sentences have been added on line 287 : “However, the specific sensitivity of 6MWD is unknown. Moreover, in COPD patients, walking distance is limited by muscle function and by respiratory capacity and it is unlikely that 6MWD could assess as well as TUG the benefit of specific balance training exercises ».

Minor comments

- Some references appear written in the text. Please revise.

We apologize for this error in formatting 2 references in the statistics paragraph. This has been corrected.

- Please review for grammar.

We carefully check for grammar mistakes throughout the text. We have corrected some and we hope the english editing is now improved.

Reviewer 2 Report

Please be so kind to provide more informations in introduction and to make more comments related to other publications in this field.

The topic is an interesting one not only for clinical purposes but also for clinical research.

Even ther is a special tool , the topic is original and add avalue to the subject area.

There is a very interesting approach of a new tool for COPD patients and very useful in clinical practice or clinical research to complete the functional evaluation of COPD patients and to predict the risk of fall.

The paper is very well written, with a strong and clear data and statistical analysis.

The conclusions are consistent with the evidence and arguments presented and nice presentation of the limitations of the study.

The article to a TUG value longer than 1 s predicts fall risk at 6- month in individuals with COPD is an interesting point of view even that is related only with a particular tool and there is no recommendation for it in GOLD COPD guidelines at this moment.

The scientific presentation is good .

The English is very well.

Author Response

We warmly thank the reviewer for his positive comments on our paper.

Round 2

Reviewer 1 Report

The authors have done a good job addressing my comments.